# Integrative Taxonomy of Two Peruvian Strains of *Brachionus plicatilis* Complex with Potential in Aquaculture

**Pedro Pablo Alonso Sánchez-Dávila** [1,*], **Giovanna Sotil** [2], **Araceli Adabache-Ortiz** [3], **Deivis Cueva** [2] **and Marcelo Silva-Briano** [3]

1    Instituto del Mar del Perú—IMARPE, DGIA, AFIA, BGOA, Esquina Gamarra y Gral. Valle s/n, Callao 07021, Peru

2    Laboratorio de Genética Molecular, Instituto del Mar del Perú—IMARPE, DGIA, Esquina Gamarra y Gral. Valle s/n, Callao 07021, Peru; gsotil@imarpe.gob.pe (G.S.); dcueva@imarpe.gob.pe (D.C.)

3    Laboratorio No. 1, de Ecología, Centro de Ciencias Básicas, Edificio 202, Departamento de Biología, Universidad Autónoma de Aguascalientes, Av. Universidad No. 940, Ciudad Universitaria, Aguascalientes 20100, Mexico; aadaba@correo.uaa.mx (A.A.-O.); msilva@correo.uaa.mx (M.S.-B.)

*    Correspondence: psanchez@imarpe.gob.pe

**Abstract:** Two Peruvian strains of the genus Brachionus were isolated from impacted coastal wetlands. With an integrative taxonomic view, we described their taxonomic status, morphological characters, productive parameters, and phylogenetic position. In the case of both strains, the relationship between biometrics and productive parameters obtained with Principal Components Analysis indicated that the lorica length was associated with longevity, progeny, egg production, and reproductive age, while the lorica width and aperture were associated with the maximum number of eggs carried. Maximum Likelihood and Bayesian Inference analysis carried out with mtDNA COI gene and rDNA ITS1 region showed that both strains were clustered in two clades with distinct phylogenetic positioning from what is currently known for *Brachionus plicatilis s.l.* One of the strains, Z010-VL, is proposed to be a subspecies of L4 (*B. paranguensis*), and the other strain, Z018-SD, is proposed as a sub species of SM2 (*B. koreanus*). In addition, 33 and 31 aquaculture production lineages are proposed, delimited by COI and concatenated COI+ITS1 sequences, respectively. Finally, this study provides new tools that enhance the traceability of the origin of each sub-species throughout the world.

**Keywords:** *Brachionus plicatilis* complex; integrative taxonomy; biometrics; lifespan; strain; aquaculture production lineages

## 1. Introduction

The taxonomy of the *Brachionus plicatilis* species complex began with the first description performed by Müller in 1786, and then other reports went on to describe the species as potentially cosmopolitan [1,2]. One of the first insights was the dominance of morphotypes according to seasonal changes, postulated by Serra and Miracle [3]. Fifteen different species have currently been postulated, but only seven formal delimited species have been described [4,5], taking into consideration the analysis of the nuclear rDNA ITS1 region. Thus, the current complex group (*B. plicatilis sensu lato*) is formed by *B. plicatilis sensu stricto* (L1), *Brachionus manjavacas* (L2), *Brachionus asplanchnoidis* (L3), *Brachionus paranguensis* (L4), *Brachionus ibericus* (SM1), *Brachionus koreanus* (SM2), and *Brachionus rotundiformis* (SS).

In Peru, studies on the taxonomy of *B. plicatilis s.l.* are scarce. Species records, ecological and morphological descriptions [6,7], as well as applied aspects [8–12] have been reported. Although there are important efforts to begin formal descriptions and to provide useful information for those wishing to experiment with multidisciplinary research with local strains, especially considering their high utility in the local aquaculture [13–16], the current taxonomic status of Peruvian strains of *B. plicatilis* is still unclear.

Biometrics is currently an instrument considered in the monitoring of resources such as fish and mollusks, which guides towards selection criteria to improve production and management. Therefore, the goal of this study is to investigate the possibility of its functional application with rotifers. The rotifers of the *B. plicatilis* species complex are the most widely cultured zooplankton used in aquaculture worldwide [17]. Hence, a taxonomic record in addition to their individual productive characteristics could prove to be a new tool to better use its potential.

In this sense, we present the results of an integrative taxonomic study of the lineage of two Peruvian strains IMP-BG Z010-VL and IMP-BG Z018-SD, from species L4 and SM2, respectively, in order to contribute to the description of the taxonomy and diversity of the species. In addition, we evaluated the correlation between productivity and biometric parameters of these *Brachionus* isolated from impacted coastal wetlands, drawing attention to this environmental issue based on individual descriptions of the life cycle of two strains and the relationship of their production parameters against three morphometric measurements.

Finally, we strengthen the hypothesis of the existence of delimited lineages in productive terms, looking to improve their traceability back to the designation of origin. Therefore, we suggest a subspecific classification to formally delimit them as aquaculture production lineages, based on morphological and molecular (mitochondrial and nuclear DNA markers) evidence.

## 2. Material and Methods

### 2.1. Sample Collection

Organisms were collected from 2 impacted coastal wetlands surrounded by desert, located in the central south of Peru (Figure 1), using a 10 μm phytoplankton net. The 2 isolated strains were coded correlatively. Strain IMP-BG Z010-VL was collected in 2009 from the Municipal wetland of the Ventanilla district, Callao (WGS84 11°52′16.11″ S; 77°08′17.88″ W), which was used as a rubbish heap by a nearby human settlement, causing it to be full of litter; while the strain IMP-BG Z018-SD was collected in 2014 from a very shallow relictual wetland near a private club in Santo Domingo, located in the Paracas district in Ica (13°51′25″ S; 76°15′11″ W). The areas of the sampling stations were registered using a Garmin GPS, model GPSMAP 60CSx (Shijr, New Taipei City, Taiwan), and the datum used in the mapping was WGS1984, displayed with the ArcGIS Desktop program, version 10.5.

The isolation of the rotifers was carried out in the laboratory to avoid undesirable protozoan. A modified pipetting technique was used for successive washes of organisms. Washes among drops of filtered and sterile seawater were performed on a glass slide, according to the protocol (unpublished) of the Instituto del Mar del Peru (IMARPE), based on Andersen [18]. Both strains are conserved ex-situ in the Germplasm Bank of Aquatic Organisms of IMARPE (http://www.imarpe.pe/imarpe/index2.php?id_seccion=I0170050400000000000000, accessed on 10 December 2021), in batch culture. Samples from different batches and from different years were selected for morphological and molecular characterization.

### 2.2. Culture Conditions, Morphometry, and Parameter Evaluation

The isolated organisms were cultured in 100 mL beakers at a density of 5 rot/mL with seawater filtered at 0.22 μm and sterilized in an autoclave. The experiments were carried out in 2 plastic 48-well culture plates, with one rotifer of each strain per well. Then, the F1 obtained the day after placing the F0 in each well was used to start the lifespan experiment. Both cultures were performed under controlled conditions in a Torrey climate chamber model R-14AI, with 14:10 h photoperiod, a temperature of 24 ± 1 °C, with water changes every 7 to 10 days, dissolved oxygen from 6.2 to 3.5 mg/L and pH from 8 to 6. The salinity was adapted in order to provide the best fit for each strain, which was observed a priori, and hence was considered 35‰ for IMP-BG Z010-VL and 25‰ for IMP-BG Z018-SD. Organisms were fed with *Nannochloris* sp.

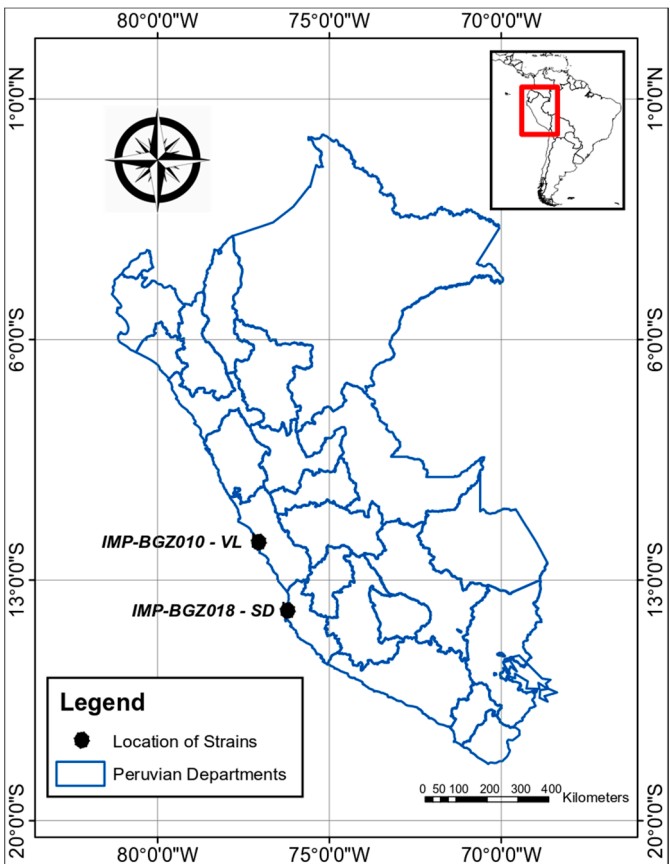

**Figure 1.** Map indicating (with black dots) the origin of the Peruvian strains, collected from Ventanilla-Callao (IMP-BG Z010-VL) and Santo Domingo-Ica (IMP-BG18-SD).

Currently, worldwide, there are no taxonomic keys to describe all the species from this species complex; consequently, 10 parthenogenetic females of each native strain were obtained from cultures and fixed in 2% formalin. These organisms were then observed under an optical microscope to describe the morphological characters used to compare and classify the *Brachionus* complex, such as the presence of gastric glands, the shape of the dorsal sinus, pores, lateral antennas, form of the eye, and the shape of the upper part of the lorica. A group of organisms from each strain was sent to the University of Aguascalientes in Mexico for the surface electronic microscopic (SEM) analysis. For this, trophi were removed according to the methodology of Segers et al. [19] and mounted on glass slides. Observations were made with SEM JEOL 5900 LV, and habit images were taken according to Silva-Briano et al. [20].

Nine morphometric character measurements of the cultivated rotifers were performed in the lapse of 3 years, using a microscope Leica DM1000 LED (Wetzlar, Hesse, Germany), with a 3 Mpxs image capture CMOS DFC290 HD (Wetzlar, Hesse, Germany) and the Leica Application Suite v. 4.10.0 software. Measurements of lorica length, distance between lateral spines, lorica width, the distance between central spines and dorsal sinus depth, the distance between central and medial spines, and medial spine length (indicated in Figure 2, from "a" to "g", respectively) were selected on the basis of Fu et al. [21]. Additionally, the characters: head aperture (h), and lateral spine length (i), were selected according to Ciros-Pérez et al. [22].

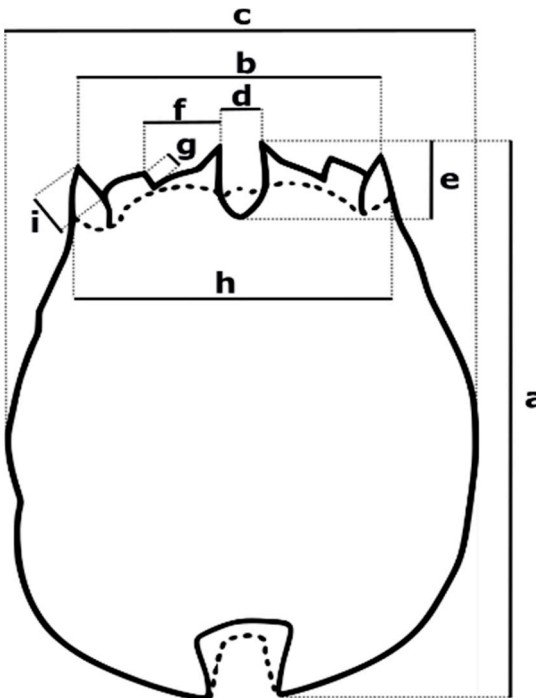

**Figure 2.** Morphometric measurements considered for the Peruvian strain descriptions. (a) lorica length, (b) distance between lateral spines—DbLS or lorica aperture, (c) lorica width, (d) distance between central spines, (e) dorsal sinus depth, (f) distance between central and medial spines, (g) medial spine length, (h) head aperture, (i) lateral spine length. Characters from (a–g) were selected based on Fu et al. [21], while (h,i) were selected considering Ciros-Pérez et al. [22] recommendations.

Initially, 5 morphometric characters ("a", "b", "c", "d" and "e") of 48 parthenogenetic females were measured at the end of each rotifer's lifespan. Moreover, from the small-scale culture, the individual production parameters from each rotifer were registered by removing the newly hatched rotifer from the plate with a small, very fine-tipped, heat-treated Pasteur pipette and replenishing the removed aliquot. We registered longevity (number of days until death), progeny (number of offspring), egg production (number of viable and non-viable eggs), maximum load of eggs (the maximum number of eggs carried by a female during her lifetime), pre-reproductive, reproductive, and post-reproductive age (periods before, during and after the reproductive stage, respectively). This production data were associated with the biometric parameters "a", "b," and "c", by the Principal Component Analysis (PCA) using the PRIMER-e version 5.0 (PRIMER-E Ltd., Plymouth, UK; https://www.primer-e.com/, accessed on 10 December 2021) software. To describe reliable size confidence intervals, we measured 5 morphometric characters (from "a" to "e") in 120 parthenogenetic females (including the previous 48 rotifers considered for PCA) of IMP-BG Z010-VL (up to 48 measures for "d" and "e"), and in 125 rotifers of IMP-BG Z018-SD.

In addition, the 9 parameters were randomly measured, regardless of age, in other 60 parthenogenetic females of both strains (IMP-BG Z010-VL and IMP-BG Z018-SD) isolated in this study and compared against 60 rotifers of 2 reference strains maintained in culture *B. plicatilis* s.s. L-size and *B. rotundiformis* SS-size. These last 240 measurements were used to statistically compare their morphology by Canonical Discriminant Analysis (CDA) using the IBM SPSS Statistics software for Windows Inc. version 22 (IBM Corp. Released, 2013 (Armonk, NY, USA); https://www.ibm.com/analytics/spss-statistics-software, accessed on 10 December 2021). The functions were obtained by stepwise discriminant analysis performed on each lorica measurement from 4 strains, and the data were transformed considering the natural logarithm (Ln).



### 2.3. Molecular Analysis

For both strains, organisms from parthenogenetic-originated monocultures were collected, and DNA extraction was conducted from each individual, as well as in pools, in 0.2 mL microtubes, using the HotSHOT method based on Montero-Pau et al. [23], with slight modifications in alkaline volume (25 µL) and neutralizing (25 µL) solutions, and with incubation on an Eppendorf MixMate microplate shaker at 800 rpm, 95 °C for 40 min. A small number of males was obtained randomly during the measurements of the females and also considered for the molecular analysis.

For species identification, the mtDNA COI gene and the nuclear rDNA ITS1 region were analyzed. The COI gene was amplified using primers ZplankF1_t1/ZplankR1_t1, while ITS1 with primers III/VIII, indicated in Table 1. PCR reactions were performed using the HotStartTaq Plus Master Mix kit (QIAGEN), with final concentrations of 0.2 µM of each primer, 2 mM $MgCl_2$, and 1.5–3 µL of template DNA, in 10 µL of final reaction volume. The thermal cycling conditions considered were an initial denaturation of 95 °C for 5 min, followed by 36 cycles of 95 °C for 40 s, 45 °C (COI) or 54 °C (ITS1) for 50 s, and 72 °C for 1 min (COI) or 45 s (ITS1), with a final extension of 72 °C for 7 min. All reactions were evaluated by electrophoresis in 1% agarose gels, amplified products with the expected size were purified with the AccuPrepPCR Purification kit (Bioneer), and bidirectionally sequenced in an ABI 3500 genetic analyzer (Applied Biosystems Inc. Foster City, CA, USA). Electropherograms were manually edited using Chromas 2.6.6 (Technelysium Pty Ltd., South Brisbane, QLD, Australia), sequences were aligned using MUSCLE [24] from MEGA 7.026 [25] and trimmed to 683 bp for COI sequences and 557 bp (VL 543 bp plus indels) for ITS1. All consensus sequences (*n* = 9 for each marker and strain) obtained in this study were registered in GenBank, with accession numbers for strains IMP-BG-Z018-SD (COI: MK534737 and MZ662909-MZ662916; ITS1: MZ569584 and MZ695037-MZ695044); and IMP-BG-Z010-VL (COI: MK534738 and MZ662901-MZ662908; ITS1: MZ569507 and MZ695046-MZ695053) detailed in Table S1. In addition, some isolates of reference strains used in this study were selected, and the COI gene was sequenced for the confirmation of species *B. plicatilis* s.s. L1-size (accession numbers OL700039–OL700040) and *B. rotundiformis* SS-size (accession numbers OL700041-OL700042).

**Table 1.** Primers used for the amplification of mtDNA COI gene and rDNA ITS1 region of two Peruvian strains (IMP-BG-Z018-SD and IMP-BG-Z018-VL) of genus *Brachionus* isolated from impacted coastal wetlands. (*) indicates primers used for COI gene sequencing.

| Marker | Primer | Sequence (5′-3′) | Size (pb) | Reference |
|--------|--------|------------------|-----------|-----------|
| COI | ZplankF1_t1 | TGTAAAACGACGGCCAGTTCTASWAATCATAARGATATTGG | ~700 | |
| | ZplankR1_t1 | CAGGAAACAGCTATGACTTCAGGRTGRCCRAARAATCA | | [26] |
| | * M13F | TGTAAAACGACGGCCAGT | | |
| | * M13R | CAGGAAACAGCTATGAC | | [27] |
| ITS1 | III | CACACCGCCCGTCGCTACTACCGATTG | ~560 | |
| | VIII | GTGCGTTCGAAGTGTCGATGATCAA | | [28] |

For comparative purposes, all COI and ITS1 sequences of *Brachionus plicatilis s.l.*, available in GenBank and BOLD databases were retrieved (avoiding the selection of misidentified sequences, with a small number of bp, a high number of gaps, or ambiguous nucleotide specifications) and aligned with those obtained in this study. A total of 256 of COI (569 bp), 197 of ITS1 (370 bp) and 197 concatenated COI+ITS1 sequences (948 bp). were analyzed. Each concatenated sequence was derived from the same organism. Parameters, including nucleotide composition, the conserved, variable, and parsimony informative (PI) sites, were calculated in MEGA 7.026 [25]. In addition, *p*-distances were calculated for all pairwise comparisons of taxa, with 1000 bootstrap replicates.

The phylogenetic relationship was reconstructed using the Maximum Likelihood (ML) and Bayesian Inference (BI) methods, with COI and COI+ITS1 sequences. *Brachionus calyciflorus* (KC431011 for COI, and KC431009 for ITS1) from Brazil was included as an outgroup.

For ML, the best substitution models were determined using the SMS algorithm [29] with optimized frequency balance and 1000 permutations. The ML analysis was performed with PHYML 3.0 [30], considering GTR + G (1.402) + I (0.555) for COI, and GTR + G (0.656) + I (0.465) for COI+ITS1. The BI analysis was carried out with MrBayes v3.2.6 x86 [31]; the Markov Chain Monte Carlo (MCMC) algorithm was run for COI, 16 million generations and 10 million for COI+ITS, and trees were sampled at intervals of 1000 generations. The first 25% of the generations (4000 trees for COI and 2500 for COI+ITS) were discarded as the burn-in phase, and a consensus tree was constructed summarizing the branching patterns of the remaining trees (12,000 for COI and 7500 for COI+ITS) and a majority rule equal to 50%. The convergence diagnostic showed all effective sample size (ESS) values greater than 1000 and the average potential scale reduction factor (PSRF+) parameters equal to 1. For the COI-based tree, the average standard deviation of the split frequencies was 0.005477, and the maximum was 0.063693. For the COI+ITS-based tree, the average standard deviation of the split frequencies was 0.005832 and the maximum was 0.052696.

## 3. Results

### 3.1. Taxonomy

- Strain IMP-BG Z010-VL
- Class Eurotatoria De Ridder, 1957
- Subclass Monogononta Plate, 1889
- Superorder Pseudotrocha Kutikova, 1970
- Order Ploima Hudson & Gosse, 1886
- Family Brachionidae Ehrenberg, 1838
- Genus *Brachionus* Pallas, 1766
- Species *Brachionus paranguensis* Guerrero-Jiménez, 2019
- Sub species *Brachionus paranguensis ventanillensis* subsp. nov.

- Strain IMP-BG Z018-SD
- Class Eurotatoria De Ridder, 1957
- Subclass Monogononta Plate, 1889
- Superorder Pseudotrocha Kutikova, 1970
- Order Ploima Hudson & Gosse, 1886
- Family Brachionidae Ehrenberg, 1838
- Genus *Brachionus* Pallas, 1766
- Species *Brachionus koreanus* Hwang, 2013
- Sub species *Brachionus koreanus santodomingensis* subsp. nov.

### 3.2. Etymology

Based on morphological and molecular analysis, the species under study have been nominated as subspecies; although it is a new category and not yet valid, it is necessary for aquaculture traceability. Strains were *Brachionus paranguensis ventanillensis* subsp. nov. (with holotype and paratypes record MUSM-PL 2021-0031) and *Brachionus koreanus santodomingensis* subsp. nov. (with holotype and paratypes record MUSM-PL 2021-0032). Both were proposed to highlight their origins since the urban sprawl growth is increasing, and these water bodies may disappear in the next few decades. The holotypes and paratypes were deposited in the Natural History Museum of the Universidad Nacional Mayor de San Marcos.

### 3.3. Morphological Differences of Peruvian Strains

A total of 180 parthenogenetic females from IMP-BG Z010-VL and IMP-BG Z018-SD, were examined and measured to describe their morphology.

### 3.3.1. Strain IMP-BG Z010-VL

Description

Strain with morphology similar to *B. paranguensis* L4, according to Guerrero-Jiménez et al. [5]. Organisms with ventral and dorsal plates fused dorsally and laterally, one triangular reddish-brown cerebroid ocellus (Figure 3b), long sensory bristles, corona with two typical concentric rings of cilia, first on the upper part trochus, formed by four bands surrounded by cirrus and cingulum on the lower part near the dorsal antenna (Figure 3c). Dorsal margin of the lorica with three pairs of spines framing the U-shaped sinus (Figure 3a), two gastric glands (Figure 3b), and two emerging lateral antennas from a lateral pore, each one placed near the widest part of the lorica (Figure 3d). Biometric results (expressed in μm) are detailed in Tables S2–S4; while productive parameters are in Table S5.

Type Locality

The parthenogenetic females were collected from the Ventanilla municipal wetland in Callao, Perú (S. 11°52′16.11″; W. 77°08′17.88″).

Differential Diagnosis

Lorica were similar to *B. plicatilis* s.s. "L1", however, statistically, the differences were enough to place it in another group. Although it is genetically related to *B. paranguensis*, one difference was found by its inconspicuous orange peel-like surface of the lorica. The anterior ventral margin of the lorica with two pairs of rounded lobules was located on both sides of the wide sinus (Figure 3a). The inner lobes with a wider base than the outer one. The toes (spurs/pedal glands) emerged completely from the foot terminal (apical) part (Figure 3e), which explains why these organisms were usually found attached to the walls of beakers, particularly during any stress due to vessel changes. The egg presented a diagonal incision on the bottom (Figure 3f). In contrast to other species such as *B. plicatilis* s.s., *B. rotundiformis* and *B. koreanus* maintained in culture for morphometric measurements in this essay, Z010-VL swam very superficially until many individuals perished, trapped by the surface tension of the water.

Trophi

The basic archetype, also called "Mallei" or "Maleate type", corresponded to the genus *Brachionus*. Hollow fulcrum, short, and truncated pyramid shape. Satellites form an irregular quadrilateral and anterior processes with rough edges in ventral view (Figure 3g). Rami with two posterior asymmetrical projections, the left one ending in a point and narrower than the right one. Basifenestras were circular and similar. Unci with four long teeth and brush-shaped subuncus. Flattened manubrium in the form of hollowed planes, each with projections towards the center and three tunnel-like cavities directed at the distally and dorsally folded ends, articulation presented its processes with a half-elliptical shape. The joints of the manubrium presented, in dorsal view, rougher ornamentation with more marked lobular bifurcation than the Z018-SD strain (Figure 3g,h).

### 3.3.2. Strain IMP-BG Z018-SD

Description

Strain with morphology similar to *B. koreanus* SM2 [32]. Organisms with ventral and dorsal plates fused dorsally and laterally, one brown cerebroid ocellus, two gastric glands (Figure 4b), sensory bristles, corona with two typical concentric rings of cilia, the first one being in the upper part trochus, formed by four bands surrounded by cirrus and cingulum on the lower part near the dorsal antenna. Pear-shaped and smoothed surface of the lorica, three pairs of spines that were triangular and dissimilar in length and width; the lateral and central spines are longer than the medial one, lateral spines have sigmoid outer margins (Figure 4a). U-shaped sinus was narrower than strain IMP-BG Z010-VL (Figure 4c). Biometric results (in μm) and productive parameters are reported in Tables S2–S5.

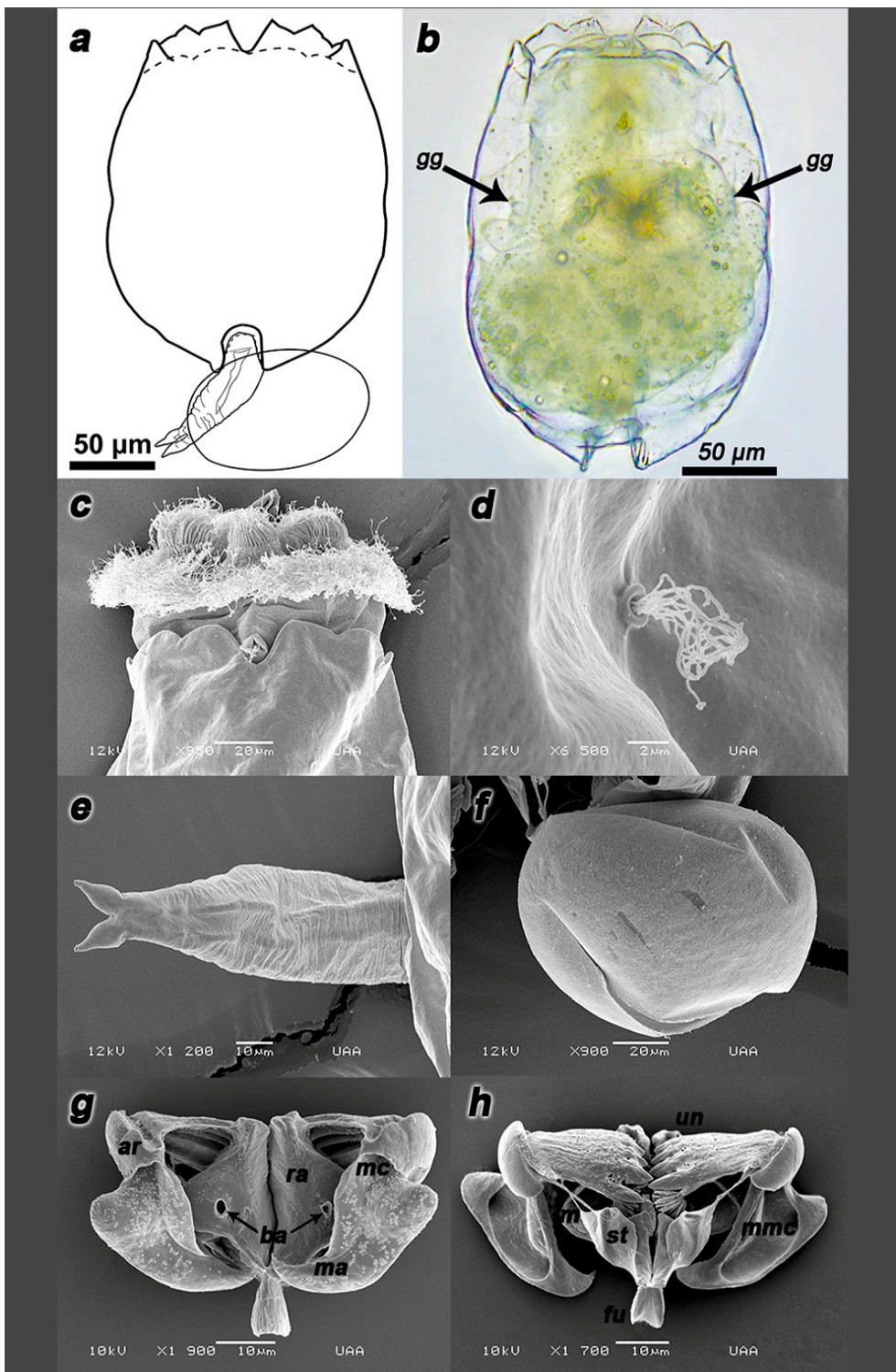

**Figure 3.** Description of the parthenogenetic female strain IMP-BG Z010-VL. Illustration of the strain (**a**) and its habitus (**b**). SEM microphotographs of the dorsal spines (**c**), pore with lateral antenna (**d**), foot (**e**), parthenogenetic egg (**f**), trophi dorsal view (**g**) and the trophi ventral view (**h**). Arrows indicate gastric gland (gg) structures. Components of trophi are indicated in cursive letters: membrane (*m*), manubrium (*ma*), manubrium middle crest (*mmc*), satellites (*st*), uncus (*un*), articulation of manubrium (*ar*), basifenestras (*ba*); fulcrum (*fu*), manubrium cavities (*mc*) and rami (*ra*). *n* = 20 rotifers.

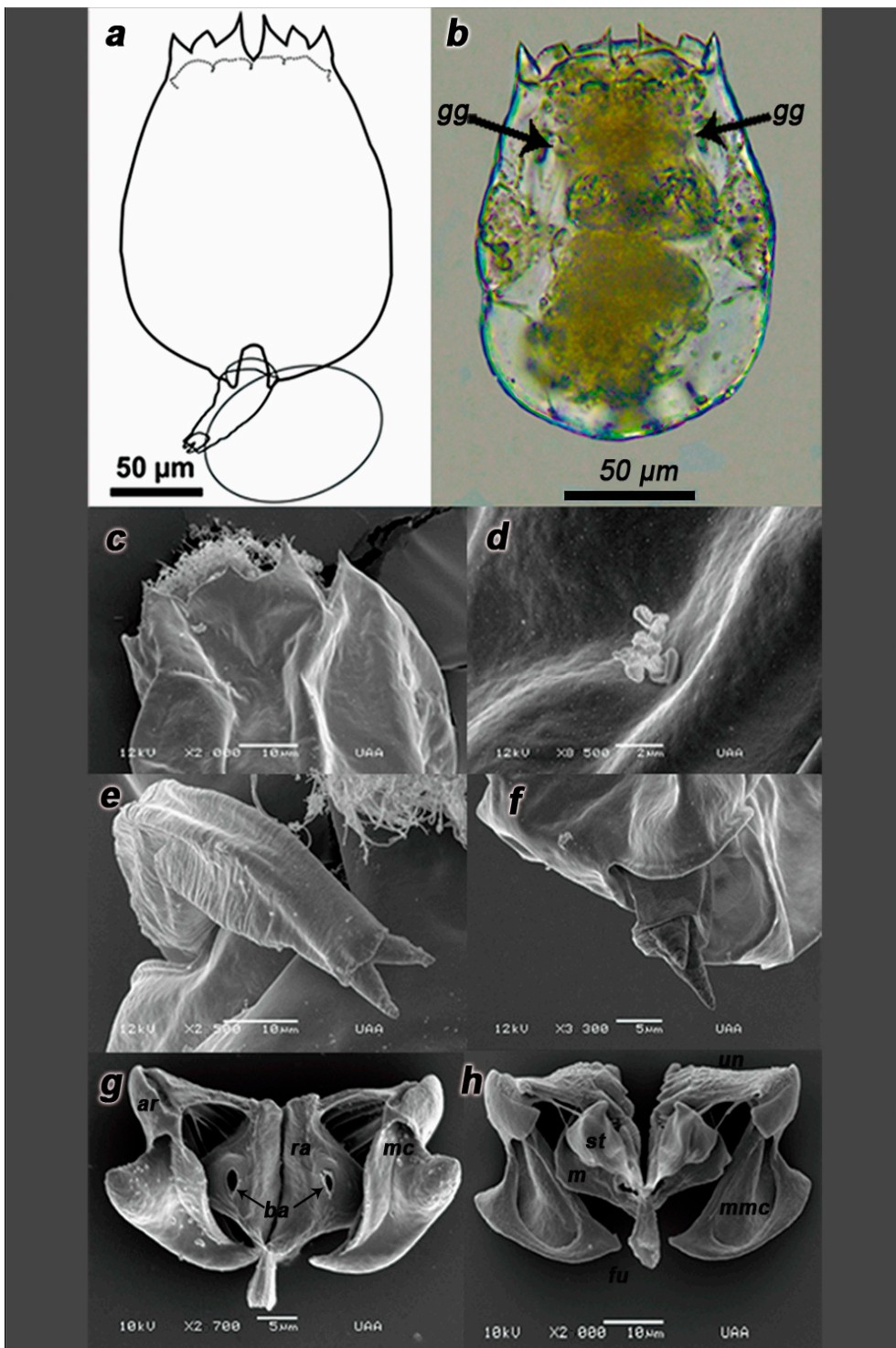

**Figure 4.** Description of the parthenogenetic female strain IMP-BG Z018-SD. Illustration of the strain (**a**), and its habitus (**b**). SEM microphotographs of the dorsal spines (**c**), pore with lateral antenna (**d**), foot (**e**), foot aperture (**f**), trophi dorsal view (**g**), and the trophi ventral view (**h**). Arrows indicate gastric gland (gg) structures. Components of trophi are indicated in cursive letters: membrane (*m*), manubrium (*ma*), manubrium middle crest (*mmc*), satellites (*st*), uncus (*un*), articulation of manubrium (*ar*), basifenestras (*ba*); fulcrum (*fu*), manubrium cavities (*mc*) and rami (*ra*). *n* = 20 rotifers.

Type Locality

The parthenogenetic females were collected from the Santo Domingo relict wetland, in Paracas, Ica, Peru (S 13°51′25″; W 76°15′11″).

Differential Diagnosis

The anterior ventral margin of the lorica with two pairs of rounded lobules was located on both sides of the slender sinus. Inner lobes showed a narrower base than the outer ones. Two emerging lateral antennas from a lateral pore, each placed near the widest part of the lorica (Figure 4d). The toes (spurs/pedal glands) did not emerge completely from the foot terminal part, which explains why these organisms were not usually found stuck to the wall of the beaker (Figure 4e,f).

Trophi

Trophi with Mallei archetype presented. Hollow fulcrum, short, and truncated pyramid-shape. Satellites form irregular quadrilateral, anterior processes with rough edges in ventral view. (Figure 4g). Rami with two posterior asymmetrical projections, the left one ending in a point to the right one, this structure was similar but thinner as opposed to IMP-BG Z010-VL. Basifenestras were irregular and oval. Uncus with four long teeth and brush-shaped subuncus. Flattened manubrium in the form of hollowed planes, each with projections towards the center and three tunnel-like cavities directed at the distally and dorsally folded ends, articulation presented its processes with an irregular triangular shape. The joints of the manubrium had, in dorsal view, smoother ornamentation with softer lobular bifurcation than the IMP-BG Z010-VL strain (Figure 4g,h).

*3.4. Relationship between Biometric and Production Parameters*

The first two principal components (PC1 and PC2) explained the 88.3% and 94.6% accumulated variation for strains IMP-BG Z010-VL and IMP-BG Z018-SD, respectively. Both PCA determined that the lorica length was positively associated with longevity, progeny, egg production, and reproductive age, while the width and aperture of the lorica (DbLS) were slightly associated with the maximum load of eggs (Figures 5 and 6).

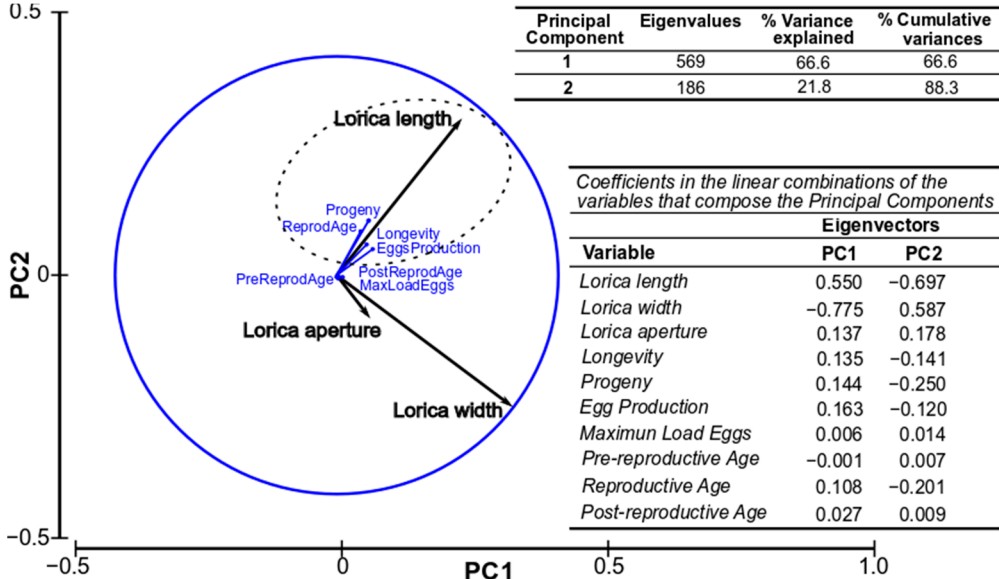

**Figure 5.** Representation of the studied production and biometric parameters of the Ventanilla Strain (Z010-VL) along the first two axes (PC1 and PC2) of the PCA. (*n* = 48 rotifers).

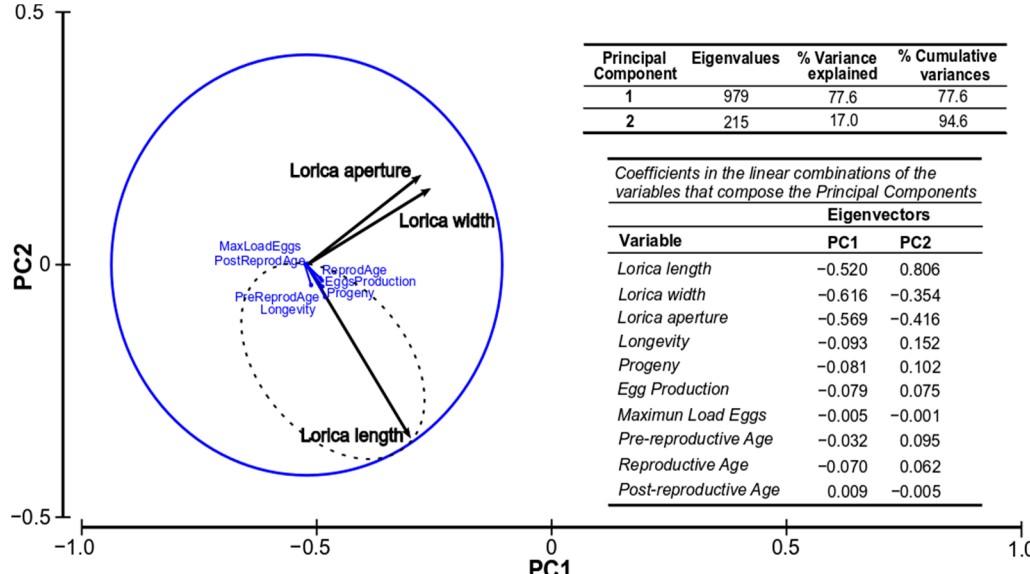

**Figure 6.** Representation of the studied production and biometric parameters of the Santo Domingo Strain (Z018-SD) along the first two axes (PC1 and PC2) of the PCA. (*n* = 48 rotifers).

## 3.5. Canonical Discriminant Analysis

According to the 96.8% (Table 2) accumulated variance from the first two canonical functions, statistical differences were reported on the four strains (*B. rotundiformis* SS-size, *B. plicatilis* s.s. L1-size, and the two strains from this study) analyzed. Only one value of *B. plicatilis* s.s. L1-size strain corresponded to IMP-BG Z010-VL. For this reason, 99.6% of original grouped cases were correctly classified. Furthermore, it is interesting to mention that strain Z010-VL was discriminated from *B. plicatilis* s.s. L1 when functions 1 and 2 were considered (Figure 7). Thus, the Brachionus strains were clearly grouped statistically in four different clusters. The higher within-group correlation coefficient in function 1 was lorica length (a), approximately duplicating the correlation values of the distance between lateral spines (b) and the head aperture (h); the same proportion of correlations gradually decreasing with dorsal sinus depth (e), lorica width (c), and lateral spine length (i), meanwhile within function 2 the distance between central spines (d), dorsal sinus depth (e), the distance between central and medial spines (f), and medial spine length (g) had the highest correlations (Table 3).

**Table 2.** Eigenvalues and variances of each function. The first two canonical discriminant functions were used in the analysis.

| Function | Eigenvalues | Variance (%) | Accumulated Variance (%) |
|:---:|:---:|:---:|:---:|
| 1 | 28.259 | 90.6 | 90.6 |
| 2 | 1.916 | 6.1 | 96.8 |
| 3 | 1.008 | 3.2 | 100.0 |

## 3.6. Molecular Taxonomy

The COI sequences obtained in this study showed a nucleotide frequency of T (42.2%), C (17.7%), A (22.3%) and G (17.9%), while for ITS1 region was T (33.2), C (16.3%), A (30.2%) and G (20.4%). Nucleotide frequencies of L4 and SM2 from different origins are detailed in Tables S6 and S7.

Considering 61 mtDNA COI sequences (570 bp) of L4 group organisms from 3 lineages of 8 countries (including those from the Peruvian strain Z010-VL), 108 PI sites were registered, and no singletons identified. In particular, 11 PI were recorded when we compared sequences from Peru and Chile (Table S4), most being due to differences between Chile

sequences. In addition, when we compared 30 sequences from the SM2 group (8 countries, including those from the Peruvian strain Z018-SD), we registered 105 variable sites (89 PI). Sequences from VL and SD Peruvian strains showed 103 variable sites (PI) and no differences within each group.

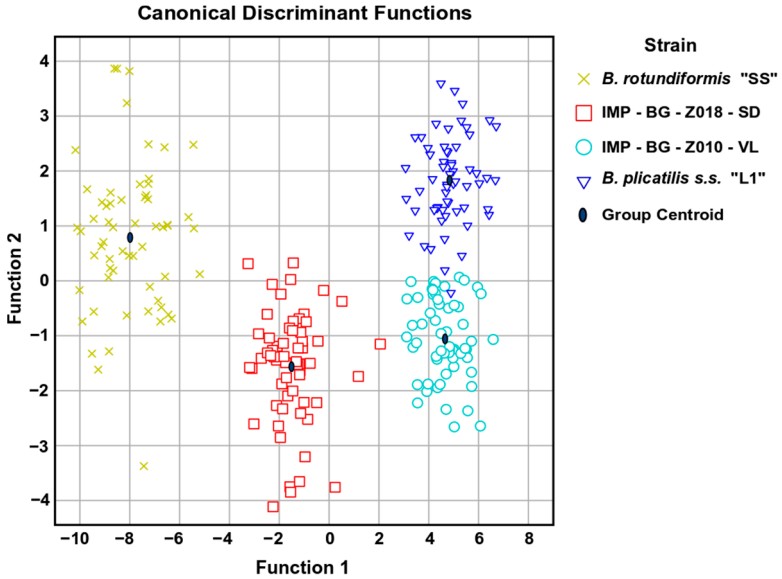

**Figure 7.** Scatter plot of each measured *Brachionus* defined by the first two canonical discriminant functions.

**Table 3.** Stepwise discriminant analyses of the body measurements for the first two canonical functions (Function 1 and Function 2). Coefficient: represents the standardized coefficient for the canonical discriminant function. Correlation: represents the pooled within-group correlation coefficient between the body measurements and the canonical discriminant function. No lorica measurement was excluded in the stepwise analyses. Measurements are presented in natural log (Ln).

| Measurement (in Ln) | Function 1 | | Function 2 | |
|---|---|---|---|---|
| | Coefficient | Correlation | Coefficient | Correlation |
| (a) Lorica length | 1.236 | 0.857 | 0.092 | 0.195 |
| (b) Lorica aperture | −0.054 | 0.43 | 0.01 | −0.08 |
| (c) Lorica width | −0.64 | 0.252 | −0.452 | 0.079 |
| (d) Distance between central spines | 0.054 | 0.022 | 0.564 | 0.69 |
| (e) Dorsal sinus depth | −0.131 | 0.266 | 0.255 | 0.542 |
| (f) Distance between central and medial spines | 0.002 | −0.015 | −0.232 | 0.498 |
| (g) Medial spine length | −0.26 | 0.112 | 0.58 | 0.46 |
| (h) Head aperture | −0.122 | 0.432 | 0.28 | −0.034 |
| (i) Lateral spine length | 0.539 | 0.237 | −0.038 | 0.059 |

On the other hand, less variability was observed when we compared ITS1 sequences from L4 and SM2 groups of different countries (along 334 positions including indels), Along 44 sequences of L4 group (from 6 countries), only one PI site (in position 328 nt, from Chile) was observed, while along 30 sequences of SM2 group (8 countries), we recorded 7 variable sites (5 PI). In addition, 51 PI sites were observed when we compared all sequences from VL and SD from L4 and SM2, respectively; and no differences were observed comparing sequences within each Peruvian strain.

Generally, higher values of uncorrected $p$-distances were registered using COI gene sequences than with ITS1. Genetic distances between COI sequences of IMP-BG Z010-VL and the L4 complex ranged from 0.9 to 13.1%, while no differences were observed with ITS1. The lowest distances were observed when comparing this Peruvian strain with sequences from Chile (0.9–1.1%), which also were grouped in the same clade after phylogenetic

reconstruction; and above 10% distance was observed compared with North America, Australasia, and Europe. IMP-BG Z010-VL and the Mexican holotype *B. paranguensis* showed 10.3% of genetic distance (Table 4). On the other hand, comparing IMP-BG Z018-SD against the other countries of complex SM2, *p*-distances ranged from 4.5 to 14.6% with COI, and from 0.3 to 1.2% with ITS1. Less genetic distance was observed comparing COI sequences of this Peruvian strain with Spain and USA (4.5–6%); whereas higher *p*-distances (up to 12–14%) were observed with Asia, the Caribbean, and Europe. After phylogenetic reconstruction, IMP-BG Z018-SD sequences showed high similarity with those from Spain but ordered in an independent clade and disrupting with North America, Asia, the Caribbean, and Europe. The IMP-BG Z018-SD strain and the Korean holotype *B. koreanus* showed 6.2–12.2% (COI) and 0.9% (ITS1) of genetic distances (Table 5).

**Table 4.** Uncorrected *p*-distance between strain IMP-BG Z010-VL and other L4 haplotypes from different countries, calculated using COI (569 bp) and ITS1 (371 bp) regions. * Corresponds to the holotype *B. paranguensis*.

| Marker | Origin of Haplotypes | | | | | |
|---|---|---|---|---|---|---|
| | (This Study) | Chile | * Mexico | China–USA | Australia–Japan–USA | France |
| COI | Peru (Z010-VL) | 0.009–0.011 | 0.103 | 0.103 | 0.131 | 0.131 |
| ITS1 | Peru (Z010-VL) | 0 | 0 | 0 | 0 | 0 |

**Table 5.** Uncorrected *p*-distance between strain IMP-BG Z018-VL and other SM2 haplotypes from different countries, calculated using COI (569 bp) and ITS1 (371 bp) regions. * Corresponds to the holotype *B. koreanus*.

| Marker | Origin of Haplotypes | | | | | | |
|---|---|---|---|---|---|---|---|
| | (This Study) | Spain | USA | Philippines–* Korea | Cayman Islands | Turkey | Italy |
| COI | Peru (Z018-SD) | 0.045–0.056 | 0.056–0.060 | 0.062–0.122 | 0.060–0.124 | 0.124 | 0.146 |
| ITS1 | Peru (Z018-SD) | 0.003–0.009 | 0.009 | 0–0.003 | 0.009 | 0.003–0.009 | 0.012 |

It is important to mention that, based only on the mtDNA COI gene, one new lineage was observed for *B. rotundiformis* (SS) from Hawaii, and another new one for *Brachionus* sp. (SM3) from Iran; while with ML analysis, lineages were not clear in L1 for *Brachionus plicatilis s.s.* (Figure 8). The BI tree (Figure 9) showed the presence of two additional aquaculture production lineages, discriminating a total of 33 lineages.

The ML and BI phylogenetic analysis based on concatenated COI+ITS1 clearly showed the presence of three clades related to the morphotypes L (L1 to L4), SS and SM (SM1 to SM7). It was also possible to discriminate the presence of 31 new aquaculture production lineages with a clear clustering related to origin in most cases. In this sense, L1 was integrated by six lineages proposed for *B. plicatilis* s.s., L2 by three for *B. manjavacas*, L3 by four lineages for *B. asplanchnoidis*, L4 by three for *B. paranguensis*, SS by four lineages for *B. rotundiformis*, SM1 by one for *B. ibericus*, SM2 by four for *B. koreanus*, SM3 by two for *Brachionus* sp., SM4 by one for *Brachionus* sp., SM5 by one for *Brachionus* sp.; SM6 by one for *Brachionus* sp. and SM7 integrated by one lineage for *Brachionus* sp. (Figures 10 and 11).

The Peruvian strain IMP-BG-Z010-VL *Brachionus paranguensis ventanillensis* subsp. nov. was clustered in the L4 group, integrated by lineage from Peru-Chile (*n* = 36 sequences). In addition, another lineage of L4 was formed by Mexico, USA, and China (*n* = 11), and a third collapsed group by France, Australia, Japan, and the USA (*n* = 14). Meanwhile the strain IMP-BG-Z018–SD *Brachionus koreanus santodomingensis* subsp. nov. was clustered in SM2 forming one lineage, discriminated from the other three lineages from SM2 formed by Spain (*n* = 3), South Korea-Turkey-Italy (*n* = 5), and USA-South Korea-Spain-Philippines-Cayman Islands (*n* = 14).

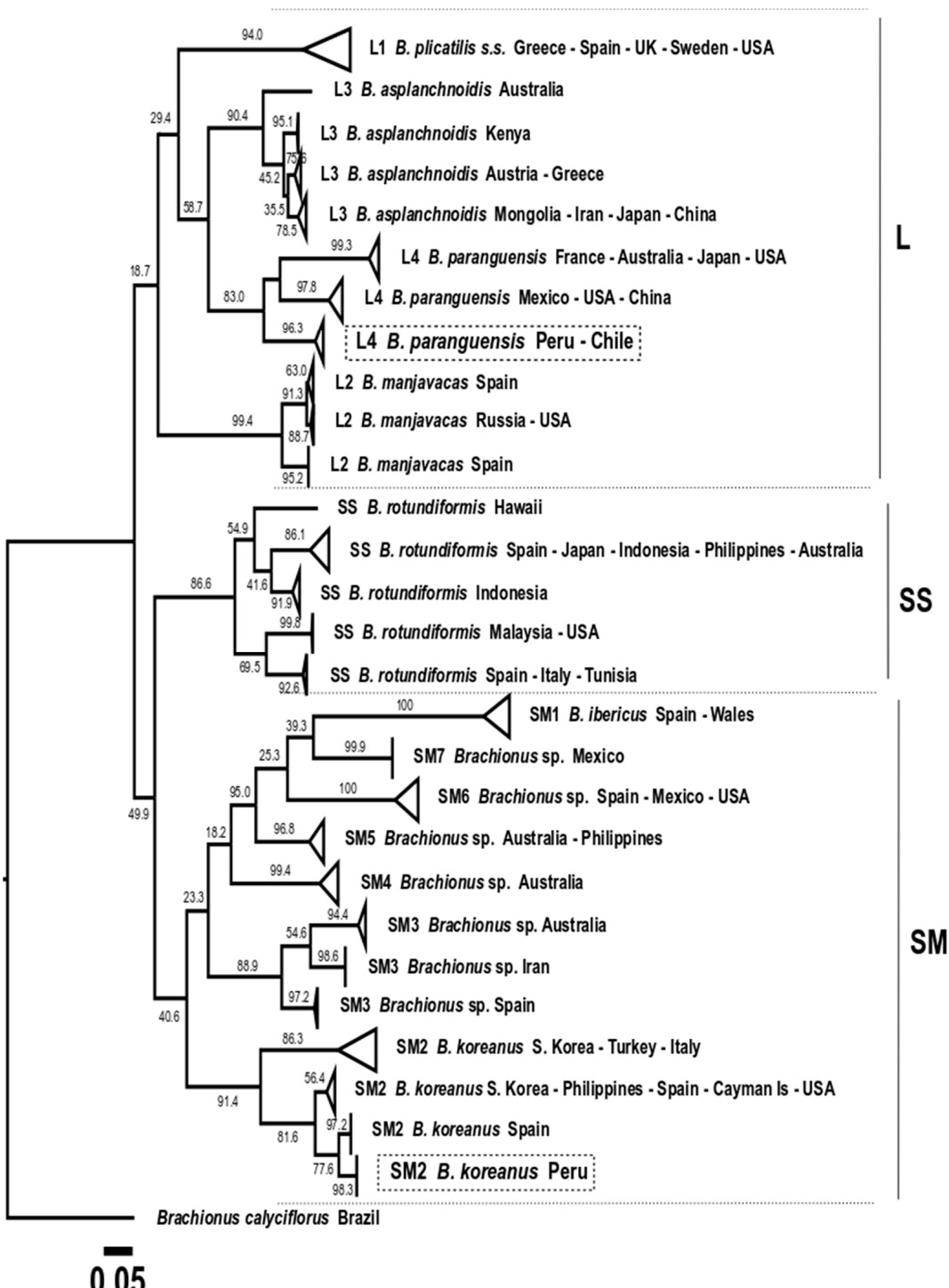

**Figure 8.** Phylogenetic tree obtained with the Maximum likelihood method based on 256 COI gene sequences (569 bp). A total of 28 production lineages were discriminated. The three morphotypes (L, SS, and SM) are indicated. Boxes in dotted lines indicate the Peruvian strain sequences obtained in this study.

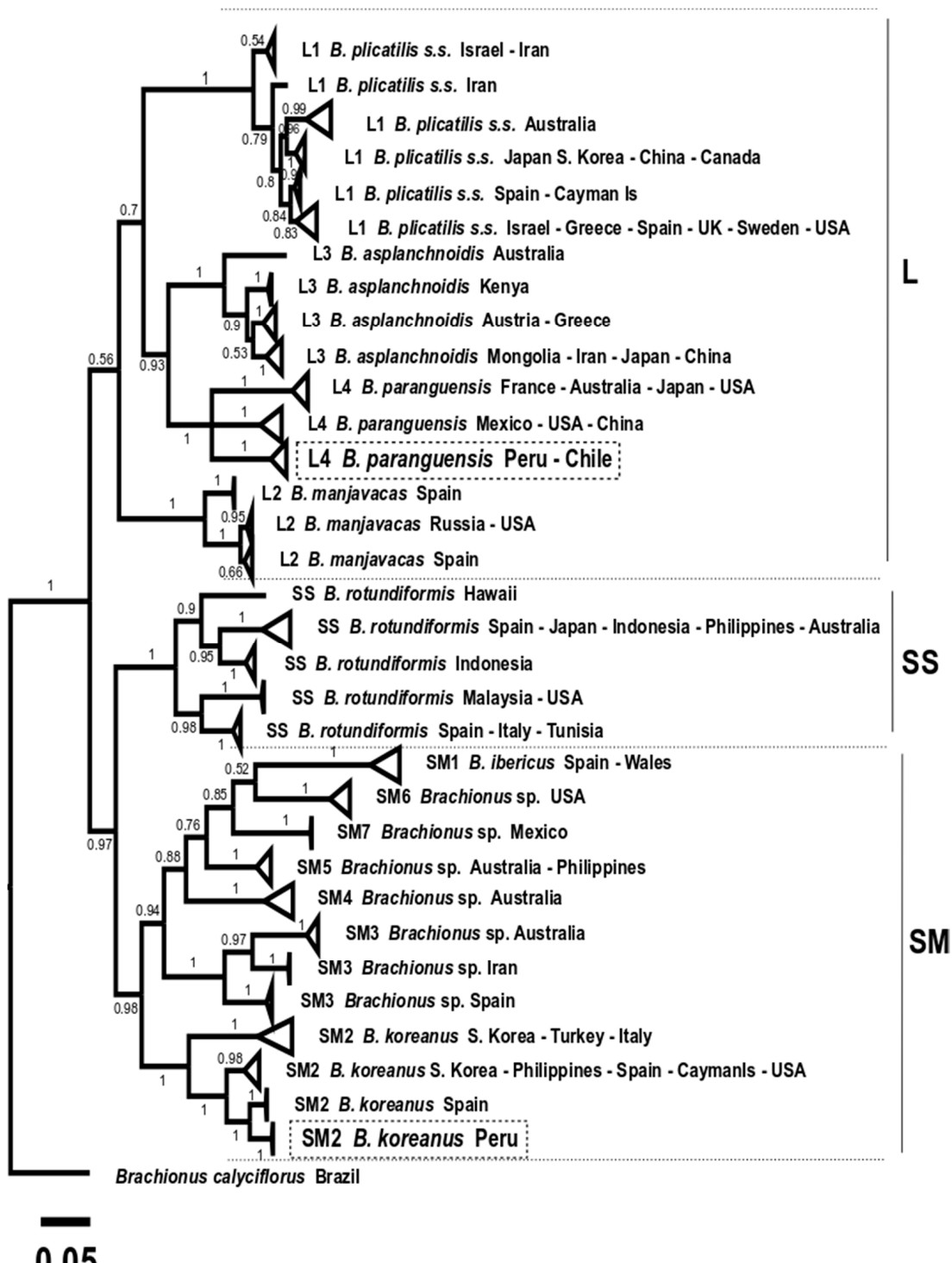

**Figure 9.** Phylogenetic tree obtained with the Bayesian Inference method based on 256 COI gene sequences (569 bp). A total of 33 production lineages are observed. The three morphotypes (L, SS, and SM) are indicated. Boxes in dotted lines indicate the Peruvian strain sequences obtained in this study.

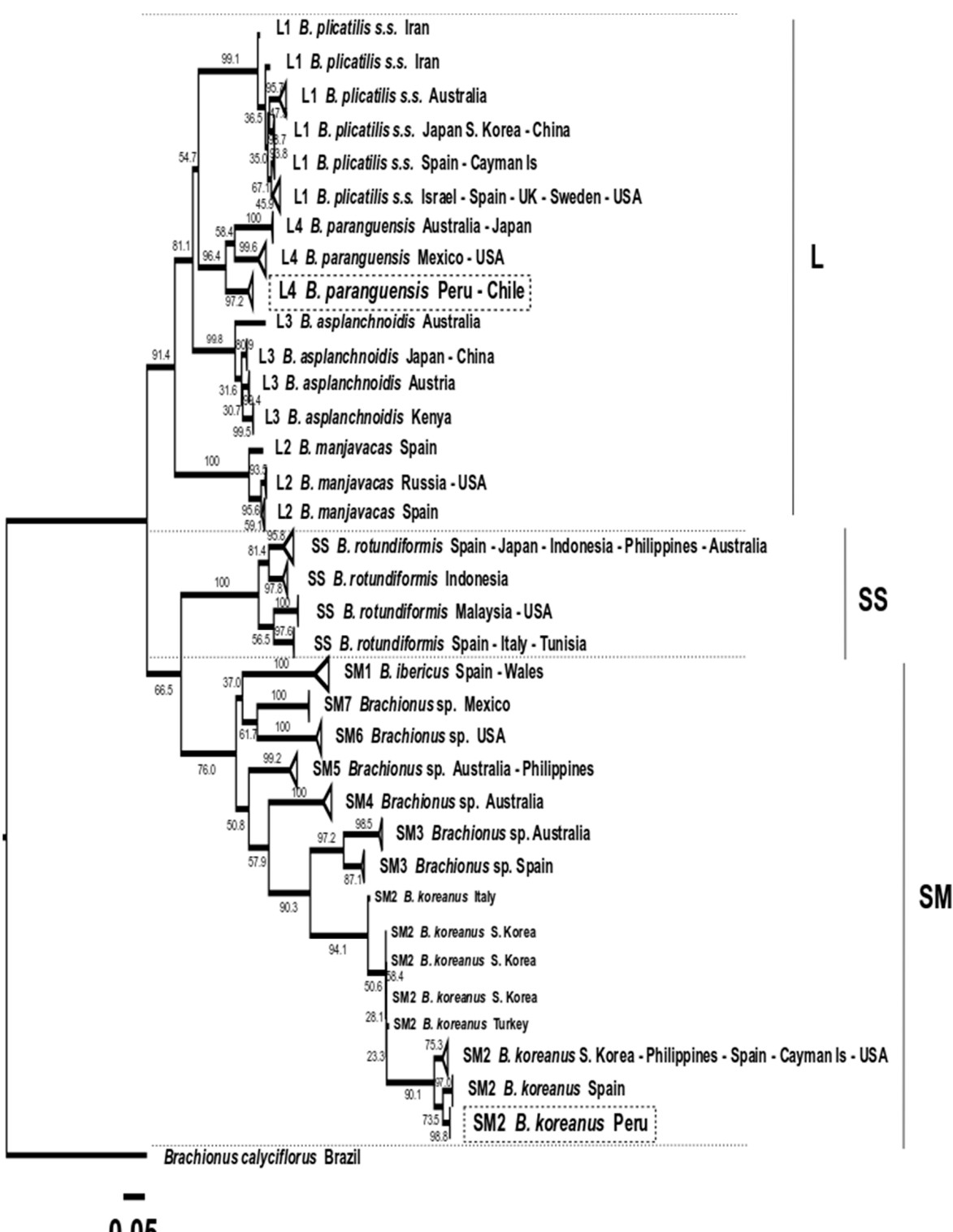

**Figure 10.** Phylogenetic tree obtained with the Maximum likelihood method based on 197 concatenated COI + ITS1 sequences, observing the discrimination of 31 production lineages. The three morphotypes (L, SS and SM) are indicated. Boxes in dotted lines indicate the Peruvian strain sequences obtained in this study.

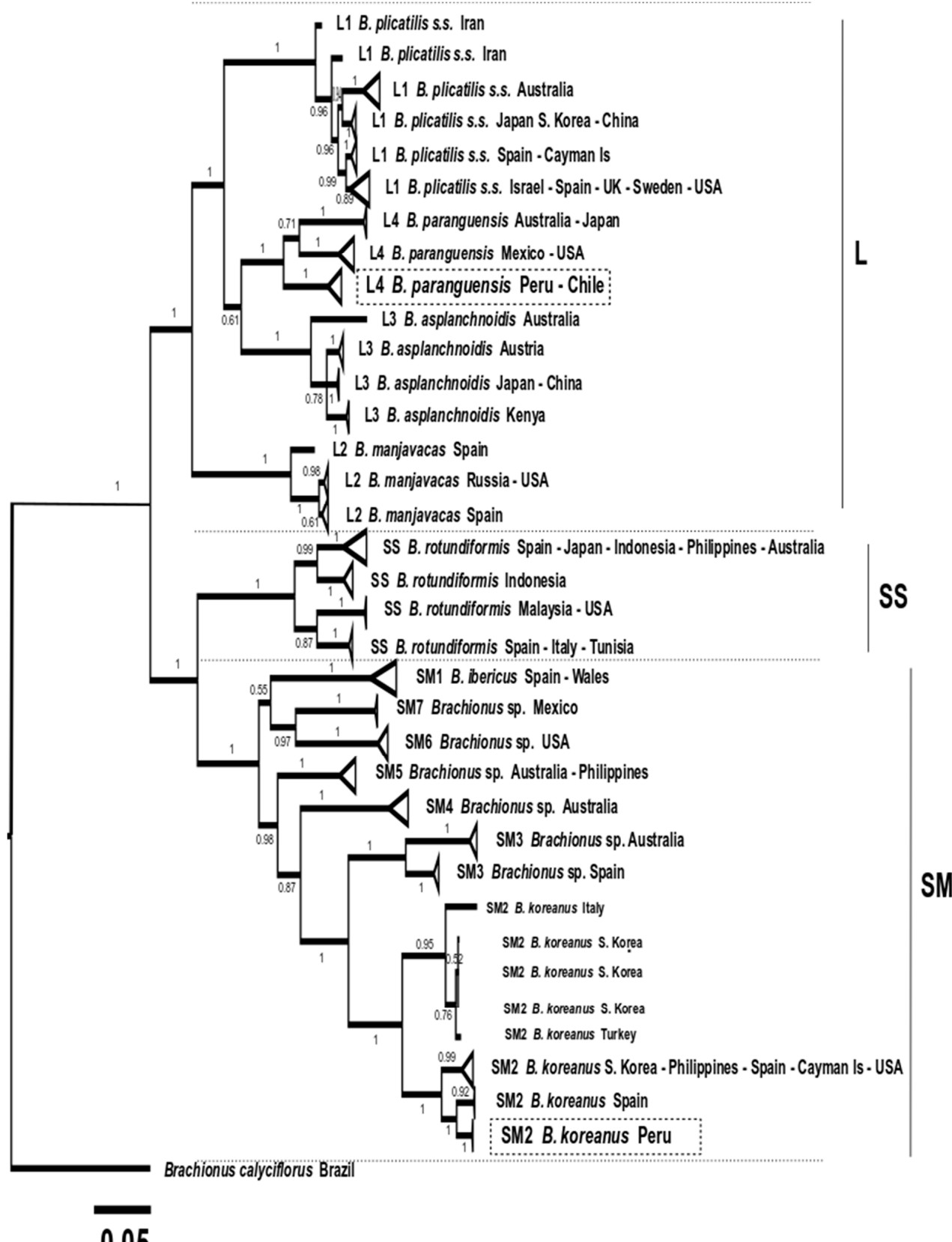

**Figure 11.** Phylogenetic tree obtained with the Bayesian Inference method based on 197 concatenated COI + ITS1 sequences (948 bp), with 31 production lineages discriminated. The three morphotypes (L, SS, and SM) are indicated. Boxes in dotted lines indicate the Peruvian strain sequences obtained in this study.

A disadvantage when analyzing lineages using concatenated COI + ITS1 was that 59 were less used than when the analysis was conducted with COI sequences, and thus samples from Hawaii (SS) and Iran (SM3) were absent, consequently showing a not well-discriminated clade. However, six subspecies groups were observed for the species *B. plicatilis* s.s.-L1.

## 4. Discussion

This study shows a morphological and molecular description of two Peruvian strains proposed here as two new subspecies, *Brachionus paranguensis ventanillensis* subsp. nov. and *B. koreanus santodomingensis* subsp. nov. These strains are considered with a potential in aquaculture, particularly in larviculture, due to the production parameters and the resilience in live food handling, also considering the habitat from which they were isolated since desert environments tend to have high levels of contamination and desiccation. In effect, taking into account a relationship between productivity and biometrics fosters value-added economic prospects to local strains that could be promoted, as well as the conservation of lineages in terms of productivity, traceability, and their intrinsic potential as new resources. Thus, the hypothesis of a delimitation subspecies is complemented with molecular (COI and COI + ITS1 regions) marker analysis. In this sense, it is important to avoid the generalization of productive parameters for all the species or genus is avoided [33] because those may be an exclusive evolutionary product by the parapatric divergence of organisms isolated from remote water bodies [34].

Morphological variability throughout ontogenetic changes is important to be considered for the characterization and selection of parameters related to productivity. Thus, contrary to different reported 72 h essays with measurements of juvenile forms in some groups [5,22,32] and even in some Asian species where they reached their maximum size after 40 h [35], here, following nine measurements recorded in four different strains of *Brachionus*, it was possible to include descriptions. In addition, with ontogenetic changes from gravid females, and also considered their standard deviation to obtain a 99% confidence interval, bearing in mind that strain Z018-SD becomes gravid the fourth or fifth day onwards (Table S5).

Due to the results obtained in this study, we propose the use of the lorica measurements ("d" to "g") as excellent parameters for successful morphometry discrimination within the genus *Brachionus*. Thus, the selected measurements (among the nine evaluated) statistically discerned four different groups (99.6% of these cases), mainly observing a taxonomic discriminant value between L1 and L4. Although the data were previously transformed to a natural logarithm to minimize the dispersion, similar differences were found without any transformation (data not shown). In addition, another important aspect to mention is that statistical differences were registered as the number of females measured randomly (up to 60) increased. Some authors reported that the somatic maintenance of females is independent of reproduction under food limitations (chronic caloric restrictions). Despite the fact that individual growth may be altered, biometrics should not be biased towards the first days of life in order to describe species but should ideally consider all measurements throughout the lifespan [36].

Phylogenetic analysis demonstrated the existence of the two new putative Peruvian subspecies, mainly based on the mtDNA COI gene analysis ($p$-distances > 3%), whilst rDNA ITS1 ($p$-distances < 1.2%) rigidly grouped them together within SM2 or L4 species. These results were consistent with previous studies, where a greater number of terminal taxa were resolved with COI-based trees compared to ITS1-based within *B. plicatilis s.l.* [4,37,38] and *B. quadridentatus* [39] and others monogonont rotifer species [40]. It has even been suggested that COI would not be appropriate for species delimitation due to the risk of an over-splitting [4], but in an intraspecific perspective, the greater variability hosted in COI sequences could reveal evolutionarily important lineages, also supported by other data and regarding economic perspectives. In fact, the biometric analysis conducted on the Peruvian strains supported the presence of statistically discrete units due to the association

of three morphological variables with production parameters. Thus, this confirmed the existence of the COI-based lineages with productive importance.

It is also noteworthy that tree topology was certainly influenced and modified depending on the molecular markers (COI or ITS1) used for the phylogenetic analysis of genus *Brachionus*, due to the low variability that ITS1 showed (variable and PI sites) compared to COI gene diversity across different *Brachionus* strains analyzed from different countries. Thus, for example, *Brachionus rotundiformis* was associated with SM group according to the COI gene analysis but was associated with L group when based on COI+ITS1. In addition, differences in the relationship between L4 and L3 were observed, COI tree topology showing them more related than with L1; while with COI+ITS1 were more related L4 and L1 than with L3. This discrepancy has also been previously reported in the phylogenetic reconstruction with ML [5], where, as in our study, higher node support values were consistently observed in the COI and COI+ITS1 tree. Conversely, Mills et al. [4] found a concordant topology between markers, supporting a closer relationship between L4 and L3 than between any of these with L1. The incongruence both in topology and level of estimated divergence between COI and ITS1, termed mitonuclear discordance, is a pervasive challenge when it is required to reconcile phylogenies singly obtained from mitochondrial and nuclear marker data sets [41]. Among the causes that generate these discrepancies are mitochondrial introgression, sex-biased dispersal, natural selection, Wolbachia-mediated genetic sweeps, incomplete lineage sorting, and unresolved phylogenetic polytomies [40,42,43], the two latter being suggested as the likely drivers of mitonuclear conflicts in some monogonont rotifer species [40].

As we mentioned previously, using only the ITS1 region may not be effective for phylogeographic studies of genus *Brachionus*, due to its remarkably conservative nature and evidence of a possible concerted evolution of these genes within the species complex [44]. Concerted evolution implies that homogenizing forces have promoted a high similarity between ITS1 sequences within the species, thus hindering the opportunity to recover bifurcation events in phylogenetic reconstructions [45,46].

*Brachionus paranguensis ventanillensis* subsp. nov.—L4 showed to have been structured in three clades (more detectable in the Bayesian than ML tree). One clade conformed by isolates from the central coast from Peru and the northern coast in Antofagasta, Chile [47], where the old migration relationship from South America to North America is located [34], continue with a volcanic area from Guanajuato, Mexico [5] and the state of Nevada, USA [4,48–50], as well as a further relationship, shown with Australasia, from Japan [4,48,51,52] and Australia [4]. We cannot explain the relationship of Chinese strains within the clade of North America or the strains from the USA within the clade of Australasia. The case of the French strains is very relevant, as well as subject to any reservations because these were also reported in Norwegian and Greek hatcheries, and their origin was mentioned by the countries where the hatcheries were located [4,53–55]. Papakostas et al. [53] investigated the species/biotype composition of *Brachionus* strains used in Europe to improve its culture conditions. One of his conclusions was that Norwegian strain "SINTEF" was the same as the Spanish strain "PL". However, our results showed these strains become a clone strain. The Norwegian strain "SINTEF" was a donation by IFREMER (France) in 1984 (Retain K.I., com.pers). They were first collected by Dr. Pourriot in 1974 in the south of Camargue [56] and were first mentioned in the work of Blanchot and Pourriot [57], naming it "*Brachionus plicatilis* GS74". Therefore, with this evidence, we considered France the country of origin for the strains SINTEF (DQ314559 and DQ314558), BEARC015, and BEARC016 (KU299273 and KU299274), and the Greek strain "K" (AM180752).

Meanwhile, *Brachionus koreanus santodomingensis* subsp. nov.—SM2 showed to have been structured in four different clades, without a clear phylogeographic relationship. One group conformed by Eurasia, Korea [32], Italy [4] and Turkey [58]; a second group was formed by Asia plus America and Europe, Korea [32], Philippines [4], Cayman

Islands [4,48], Spain [4] and USA [48]; a third group Europe, from Spain [4] and the fourth group from South America, in which Peru was included.

For *Brachionus* morphotypes analyzed in this study, we confirmed the possibility of differentiating clades related to geographic origin observed in this study based on COI and concatenated data, and we suggest including information on production parameters in the study of these groups as a future aspect for traceability of exchanges among hatcheries. Previous studies highlighted the importance of the intraspecific component in understanding the geographic patterns of biodiversity in microorganisms [59], but disentangling such geographic patterns below the species level, this sublevel could also be considered for an economic perspective when information on production parameters is added. For example, knowing the origin place of the production of lineages would improve the management of local strains by allowing traceability during their commercialization. Therefore, information on production parameters is an important aspect to be completed for the other worldwide production lineages complementing those two presented in this study in order to allow business traceability among hatcheries.

This new subspecific delimitation proposed using COI and COI+ITS1 markers could be understood, based on the economic delimitation of aquaculture production lineages, that despite a few morphological differences, this could be supported with production parameters. For example, it has been suggested to supplement taxonomy with the life cycle and the highest reproductive rate at the best salinity and temperature [60]. Those parameters must be the result of their adaptability to divergent habitats with different food, salinities, and temperature as limiting factors [61]. Kutikova [62] remarks behavioral patterns to suggest phylogeny within rotifers. However, the family *Brachionidae* preserves differences in food capture and types of body movements. Snell [63] conducted a complete revision about rotifer lifespan and reproduction, interpreting some production relationships with dietary restrictions and genetic expression.

Considering the above, a common pattern in this essay was observed in both local strains, where lorica length is associated with positive production parameters. However, this inference could be better represented with more robust statistical tests, such as CoInertia analysis (Table S8), also including more strains or subspecies. Apparently, this could happen because longer rotifers may be faster than shorter ones. This swimming speed range could be interpreted as a better chance to find food (i.e., motile algae) as well as better fitness to spread their eggs further, therefore increasing the likelihood of their progeny's survival. Whereas association of lorica width and aperture (DbLS) with a maximum load of eggs could be explained because all rotifers were measured after their complete lifespan. Thus, their bodies were influenced in their final form by these two width parameters, as carrying more eggs requires more strength to pull them, slowing them down. Korstad et al. [64] reported an inverse relationship between swimming speed and density (rotifers number per ml). In particular, swimming speed became slower in the stationary phase. In this way, we suggest that the corona needs to be stronger, and thus the trochus and cingulum must fill more space in its lorica. We have also observed that rotifers with more than two eggs had lesser hatching thereof, and these eggs were often carried joined downside the lorica near the foot. In this study, the unique case of six eggs carried by a female of *B. paranguensis ventanillensis* subsp. nov. had a single offspring, which was male and lived for 3 days.

We must also mention that both Peruvian strains showed different stress tolerance during cultures, this being another parameter for discrimination. The management of each strain must be considered separately. For instance, IMP-BG Z010-VL in a batch or clonal culture [65] tends to stick onto the beaker, and it is common to find them dead and floating on the surface due to this behavior of agglomerating very close to the waterline, causing them to become trapped by the surface tension. For this reason, it is highly recommended to gently shake the beaker occasionally to avoid the loss of this strain. The case of Z018-SD is possibly simpler, considering they tend to swim homogeneously in the water column. However, this strain is more delicate to water changes and is easily stressed due to any

other changes such as food, temperature, oxygen, or salinity. Both cases highlight first drafts of new behavioral differences among local and regional lineages, particularly in fresh tropical waters [51].

The incorporation of biometric data for the other two subspecies of *B. paranguensis* and the other three subspecies of *B. koreanus* not considered in this study is recognized as an important aspect that should be included in subsequent studies in order to strengthen this hypothesis. In addition, the lack of updated morphological taxonomic keys for the *Brachionus plicatilis* species complex is a relevant issue to new researchers and students who wish to deepen future investigations, which is why more research must be performed in order to integrate them all in one complete morphological key, backed up with molecular support, at least for the 15 putative species reported [4], also considering a need for preventative practices in uncontrolled international exchanges of rotifer strains among institutions or common sales, as they are settled with unnamed strains or without registration of origin traceability, which is unfortunately regarded as a common informal practice.

Finally, it is important to call attention to Peruvian ecological issues. The Ventanilla wetlands in the Province of Callao have serious environmental plights [66–68], as there is also a growing human settlement nearby that contributes to the large amounts of litter found in the area. In addition, other wetlands nearby, called the Regional wetlands, have been re-designed from their natural state for landscape tourism reasons. An example of this is the small colorful fish that have been introduced, with authorities further linking part of those scarce bodies of water with canals and building wooden bridges over them. The Santo Domingo wetland in the Paracas district is a complex of very shallow and little brackish water bodies (<30 cm depth $\times$ 2 m length), with a few trees and lower vegetation in the middle of a wasteland near the highway and a residential area called Santo Domingo, from which its name originates. In spite of Monogononta dormancy strategies to resist long periods without water [69,70], it is very likely that these two Peruvian coastal wetlands will disappear in the next few years, as others less known did, and in consequence, local fauna such as birds, reptiles and above all plankton and benthos invertebrates too, unless authorities start to better organize urban growth regarding the surrounding wetlands [71].

## 5. Conclusions

Morphological and molecular differences were reported against four *Brachionus* species. We point out that the COI marker is suitable for classifying hydrobiological economic resources, which can be ordered as subspecies, while the ITS marker can delimit species complex.

The IMP-BG Z010-VL strain presented resilience to daily manipulation, showing potential in aquaculture applications, and corresponded to *Brachionus paranguensis*—L4. Meanwhile, the Z018-SD strain was less resistant to manipulation, getting stressed easily, and corresponded to *Brachionus koreanus*—SM2.

*Brachionus paranguensis*—L4 showed three clades geographically differentiated (Group 1: North America; Group 2: Australasia in addition to an old formal registry from France; and Group 3: Peru and Chile); while *Brachionus koreanus*—SM2 showed four clades that were not geographically well defined (Group 1: Eurasia; Group 2: Asia, Europe and America; Group 3: Spain; and Group 4: Peru).

In both cases, the biometrical relationship with production parameters determined that the length of the lorica is positively associated with longevity, progeny, egg production, and reproductive age, while the width and lorica aperture are slightly associated with the maximum egg load.

The presence of local subspecies of *Brachionus plicatlis* complex from impacted and vulnerable environments was evaluated. We recommend further studies to support taxonomy beyond biodiversity inventories and suggest a more multidisciplinary and applied approach must be performed in order to offer more decision-making tools, such as economic added value as an instrument to support conservation.

**Supplementary Materials:** The following are available online at https://www.mdpi.com/article/10.3390/d13120671/s1, Table S1: List of GenBank accession numbers for COI and ITS1 sequences obtained in this study, from organisms of strains isolated from Peru, Table S2: Measures from parthenogenetic females of the strains IMP-BG Z010-VL and Z018-SD, Table S3: Measures from parthenogenetic females of the strains IMP-BG Z010-VL and Z018-SD used for statistical comparison (*n* = 60), Table S4: Measures from parthenogenetic females of the reference strains L (*B. plicatilis* s.s.) and SS (*B. rotundiformis*) used for statistical comparison (*n* = 60), Table S5: Record of production parameter of both parthenogenetic females of the strains IMP-BG Z010-VL and Z018-SD. Min = minimum, Max = maximum, SD = Standard deviation. The units of each parameter are indicated in parenthesis. Data obtained from the F1 with less than 24 h after hatching. (*n* = 48), Table S6: Nucleotide composition registered in mtDNA COI sequences (570 bp) of L4 and SM2 organisms from different origins, considered for phylogenetic analysis, Table S7: Nucleotide composition registered in rDNA ITS1 sequences (331 bp) of L4 and SM2 organisms from different origins, considered for phylogenetic analysis, Table S8: Summary of the outputs of the R-Studio programme. They showed low correlation expressed by Monte-Carlo Test on the sum of eigenvalues of a co-inertia analysis (RV).

**Author Contributions:** P.P.A.S.-D. conceived the idea for the project and registered morphological data. M.S.-B. and A.A.-O. performed the SEM and morphological description. G.S. performed the DNA extraction, amplification, and sequencing. P.P.A.S.-D., G.S. and D.C. analyzed the molecular data. All authors have read and agreed to the published version of the manuscript.

**Funding:** Research was partially funded by the Ministry of Production—IMARPE-DGIA (PpR Budget program 0094 "Ordenamiento y Desarrollo de la Acuicultura"). The specimens analyzed are property of the Germplasm Bank of Aquatic Organisms Germplasm Bank (BGOA for its acronym in Spanish) of the Instituto del Mar del Peru.

**Institutional Review Board Statement:** Ethical review and approval were waived for this study, due to the institutional management of ex-situ conservation of native strains with aquaculture potential. Besides, regarding Peruvian law Nº 30407. Law for the protection and welfare of animals. Article 25. Prohibitions and exceptions for the use of animals in acts of experimentation, research and teaching. Clause 3.

**Data Availability Statement:** The data presented in this study are available on request from the corresponding author. The data are not publicly available because it belongs to the Instituto del Mar del Peru.

**Acknowledgments:** The authors thank the BGOA team and the Biologist Elder Fernandez for plotting the map, and also thank Gerardo Guerrero Jiménez for his help preparing the trophi of the specimens.

**Conflicts of Interest:** The authors declare no conflict of interest.

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
