# Peer review of "Integrative Taxonomy of Two Peruvian Strains of Brachionus plicatilis Complex with Potential in Aquaculture"

_diversity, doi:10.3390/d13120671_

Round 1
Reviewer 1 Report
The present manuscript has potential for publication as it provides a large set of morphological, morphometrical and molecular data. However, the manuscript has some weak aspects which must be addressed or deleted. The following issues endorse the above view.
- The title is misleading. Authors have not provided any detailed information about the culture conditions, culture stability, egg ratios and males and/or resting eggs.
- Brachionus plicatilis complex offers two different applications in aquaculture: high growth rates for biomass production or cyst production for export (see numerous works of TW Snell). In this work neither was presented (lines 123-127)
- The supplementary table S5 presents some data on longevity offspring, production, maximum charge (eggs), pre-, post- and reproductive age. All these are misleading as all of them lack standard deviation and many of them have no units. It is not clear how the authors obtained these data without any explanation in the Methods section. What is max charge for eggs?
- The method of isolation also offers scope for doubts, because it does not say how clonal population was obtained. This could be mixed strains (lines 86-90)
- Morphometry does not automatically lead to aquacultural applications. Did the authors obtain high density cultures or high resting production?
- The taxonomic description is also faulty. For example, Brachionus paranguensis sp. nov. Guerrero-Jiménez, 2019. If the species was described two years ago, how did this taxon still retains sp. nov. status? and how long will it continue?
- Table 2 is a burden to the journal.
- Lines 595-611 are superfluous to this manuscript. Authors did not present any data on the rotifer swimming rates, feeding and filtration rates or the egg hatching. Egg carrying abilities cannot be attributed to lorica size. One of the smallest rotifer genera, Anuraeopsis can carry up to 6 parthenogenetic eggs without ill effects on swimming speed. Moreover, the interpretation presented by the authors lacks support from literature and articles were not cited to substantiate the claim.
- Certain terminology used in lines 611-622 is also misleading. Axenic cultures of rotifer have not been established in the world so far. Only J.J. Gilbert established monoxenic cultures during 1970s.
Minor errors: In many places, the genus or the species were not in italics (for example, line 549).
Author Response
Please see the attachmet.
Thank you.

Reviewer 2 Report
General Assessment (minor revision - as in essence the inclusion of a few in silico analyses is required; phylogenetic species delimination may require some time)
This is a very extensive and well-executed work in the description of two rotifer strains of the Brachionus plicatilis complex of cryptic species from Peru. The authors collected different kinds of data, from phenotypic measurements and life-history traits to genetic information on barcoding genes, and conducted a number of interesting analyses to suggest that their examined strains may be categorized as subspecies within a previously characterized cryptic species. I like the effort by the authors, and I find that the study has merit to be published in the journal Diversity. I advise the editor to provide the authors with the necessary time (more than just a few days) to address my comments listed below as this work fits well the scope and mission of Diversity.
Major Comments
- 68-73: I am very confused by this paragraph. Are the authors trying to introduce a new definition of species here? A species definition that is associated with aquaculture “productive terms”? In my opinion, this isn’t in the right direction. Common practice in B. plicatilis applied and basic research is largely based on the biological species concept with population growth and aquaculture performance largely determined by the ecological preferences of each species.
- Relationship between biometric and production parameters. I have some concerns with describing such association for such variable traits. At the very least some significance testing needs to be performed on the deduced associations. Based on the short length of eigenvectors in Figures 5 and 6, I have some concerns whether any of the examined associations is significant.
- I am not convinced that the morphological analysis is ideally suited to support the categorization of the examined strains as subspecies. Ideally the morphological comparisons need also include samples of B. paraguensis and B. koreanus other than Peru – Chile (according to Fig. 8). At the very least, the authors should acknowledge such a shortcoming in the manuscript.
- I would advise the authors to include an analysis of phylogenetic species delimitation such as GMYC and ASAP to provide additional phylogenetic insights in support of their categorization of their strains as subspecies.
Minor Comments
- 18. No need for the code information line at the Abstract.
- 29-31. Unclear sentence; please rephrase.
- 40. “breaking points” isn’t understood. Please use a different expression.
- 41-43. Mills et al. (2016) [ref 4] provided enough evidence in support of all 15 species. That some species are missing formal names does not cancel this evidence.
- 51: fix “researches”
- 56: “so why not use it with rotifers?” is too informal for scientific writing. Please rephrase.
- 62: remove ‘done’.
- 105: “Currently worldwide, there are no taxonomic keys of this species complex”. I believe that the authors should revise this statement. For instance, Michaloudi et al. 2015 and Ciros-Perez et al. 2001 (articles that the authors cite) include taxonomic keys for some of the cryptic species of the complex.
- 128-129: PRIMER-e version 5.0: add citation or access link.
- 136: How did the authors identify those reference strains, B. plicatilis s.s. and B. rotundiformis? By sequencing?
- 138: SPSS for Windows: add citation or access link.
- 153: what are the slight modifications?
- 179: “(avoiding those with errors)”. It is unclear what the authors mean. Please revise.
- 182: “concatenated COI+ITS1 sequences”. For the concatenation, both sequences need to derive from the SAME DNA sample; it doesn’t make sense otherwise to link genetic information from different DNA samples. Please clarify in the manuscript this point.
- 196: “consensus tree”. Based on which criterion the consensus tree was built? 50% majority rule? 90% majority rule? Something else?
- Figure 5 & 6 legends: Please add sample sizes (n).
- Figures 7-11. No need to have that both ML and BI trees per case in the main manuscript. One set of trees (ML or BI) can be included as supplementary information.
Round 2
Reviewer 1 Report
Please replace monoxenic to clonal cultures. Except Gilbert no one has ever produced monoxenic cultures of rotifers.
Reviewer 2 Report
General Assessment (major revision)
This is an improved manuscript. A few remaining comments from me and onto final acceptance.
Important remaining point: I strongly advice the authors to have their final version checked for English language by a native English speaker colleague or a paid service. This is such a hard work that it’s a shame to allow negative impressions from grammar mistakes that can be easily corrected by a native English speaker otherwise.
Regarding my previous major comment #1 (reviewer 2), I suggest the following:
- Previous major comment #1. I don’t agree, but I respect the opinion of the authors that an “economical” sub-classification of those organisms may be proposed and “judged” by the scientific community. However, this is a topic of the discussion section. I thus strongly advise the authors to transfer this paragraph from the introduction to the discussion section and enrich it with some of the arguments they present in the response letter (and with support from the results presented in this present work).
Minor Comments
- Define MI and BI on first use in the text and use full names in the abstract.
- “different from all others known from Brachionus plicatilis s.l. So, Z010-VL is proposed to be a sub species of L4 (B. paranguensis) and Z018-SD a sub species of SM2 (B. koreanus).”. You may change it to: “with distinct phylogenetic positioning from what is currently known for Brachionus plicatilis s.l. One of the strains, Z010-VL, is proposed to be a sub species of L4 (B. paranguensis), and the other strain, Z018-SD, is proposed as a sub species of SM2 (B. koreanus).”
- “Also, 33 and 31 aquaculture production lineages were proposed”. Change “were” to “are”.
- The new sentence in lines 26-28 is unclear. Trying to understand the meaning of it, I think it can be also removed.
- “so therefore we could investigate the possibility of its functional application with rotifers.”. You may change this to: “Thus, a goal of this study is to investigate the possibility of its functional application with rotifers.”
- Materials & Methods. Line 84. “The rotifers isolation was carried out in the laboratory”. Change to: “Isolation of the rotifers was carried out in the laboratory”.
- 622: “this could happen”.
- 680: Change “rusticity” to “resilience”.
